

# Body mass index and attention bias of food cues in women: a mediation model of body weight dissatisfaction

Aibao Zhou[1,*], Pei Xie[1,*], Md Zahir Ahmed[1], Mary C. Jobe[2] and Oli Ahmed[3]

[1] Department of Psychology, The Northwest Normal University, Lanzhou, China
[2] Department of Psychological and Brain Sciences, The George Washington University, Washington, D. C., United States of America
[3] Department of Psychology, University of Chittagong, Chattogram, Bangladesh
[*] These authors contributed equally to this work.

## ABSTRACT

Food attention bias could be used to indicate diet-related diseases in individuals with obesity. The purpose of this study is to explore the relationship between body mass index (BMI) and food attention bias, and the mediating role of body weight dissatisfaction (BWD) on this relationship in women. Seventy-five participants were recruited to complete a visual dot task with eye tracking. The results showed that BMI would positively predict response latency and duration bias on high-calorie foods; the relationship between BMI and response latency of high-calorie food was a complete mediation of BWD; the relationship between BMI and duration bias of high-calorie food was a complete mediation of BWD; and BWD positively predicts response latency and duration bias on high-calorie foods. These findings suggest a positive relationship between BMI and food attention bias, and the effect of a complete mediation of BWD in women.

## INTRODUCTION

In contemporary society, obesity has gradually and increasingly endangered people's health. Since the 1970s, obesity has tripled globally (*World Health Organization, 2021*). In 2015 along, a high body mass index (BMI) accounted for 4.0 million deaths worldwide, nearly 40% of which occurred in persons who were not classified as having an obese BMI (*Afshin et al., 2017*). A high BMI has also been found to increase the risk and severity of chronic diseases (*e.g.*, cardiovascular disease, diabetes, cancers) (*Singh et al., 2013*; *World Health Organization, 2021*). In general, there is a strong association between body weight and food. Compared to normal weight people, those with high BMI showed an increased attentional bias to food cues (*Hendrikse et al., 2015*). Food attention bias is defined as the tendency to focus on food rather than neutral information (*Thomas et al., 2015*). It as an objective index of motivational state can shed light on the cognitive mechanisms and role of correlative influencing factors of eating behavior (*Hepworth et al., 2010*). Food attention

Corresponding author
Pei Xie, 2020104074@nwnu.edu.cn, xiepeipeipsy@163.com

bias could predict subsequent food intake, type of intake, etc. (*Werthmann et al., 2011*). However, *Stamataki et al. (2019)* suggested that BMI could not alter attention bias to food.

A recent meta-analysis by *Hagan et al. (2020)* was conducted a dot-probe (automatic attention, maintained attention) and an emotional Stroop to test whether people who are overweight or obese would showed greater attention bias to food compared to people with a normal weight. Meta-analytic results showed that people who are overweight or obese did not differ from those with a normal weight on attention bias to food cues. According to cognitive-behavioral theories, the food cues bias in people with high BMI could reflect activation of dysfunctional food schemas, which showed greater attention to food (*Kemps, Tiggemann & Hollitt, 2014*; *Williamson et al., 2004*). *Kaisari et al. (2019)* adopted two attention tasks to investigate whether attention bias for food cues varies between people who are overweight or obese and people who are a normal weight. Their study determined whether the bias predicted food intake in a taste test and weight change over one-year. The results showed that people who were overweight or obese manifested a greater top-down attention bias towards food cues, compared to people with a normal weight; top-down attention bias complies people's longer-term cognitive strategies, for example, biasing attention colored spots which great chance be a food if they are hungry (*Connor, Egeth & Yantis, 2004*), towards food cues also predicted weight change over one-year. People with morbid obesity showed that dopamine D2 receptors decreased and could develop resistance to leptin, leading to forced eating. This overconsumption promotes the increased release of endogenous opiates, increasing appetite which determines weight gain and obesity (*Campana et al., 2019*). Additionally, *Ravichandran et al. (2021)* found that individuals with high BMI who had a food addiction showed greater connectivity between brainstem and reward regions compared to individuals with no food addiction; and BMI was positively associated with connectivity between the brainstem and central autonomic network in individuals with no food addiction.

Recently, a study found that women with high BMI manifest more emotional overeating than males with a high BMI among obese people (*Ravichandran et al., 2021*). Although eating disorders could occur in both males and females, females tend to display greater associations between food addiction and BMI than males (*Hallam et al., 2016*; *Şanlier, Türközü & Toka, 2016*). Surrounded by different kinds of body information from mass media, females could easily be influenced by thin and attractive media images (*Te'eni-Harari & Eyal, 2015*). According to sociocultural theory of the tripartite influence model, sociocultural factors like family, peers, and media could affect satisfaction with physical appearance, it is through sociocultural factors effect appearance comparison and internalization of the thin idea to effect satisfaction with physical appearance (*Thompson et al., 1999*). *Shen et al. (2022)* found excessive discussion and valuation of body weight among peers caused body dissatisfaction and excessive comparison and of appearance also promotes body dissatisfaction among Chinese college students. Therefore, when women have a preoccupation with body appearance or weight and hold high standards for their body, but their real body weight doesn't match the standards, this may create body dissatisfaction. Previous research showed that body satisfaction for women is usually related to weight, which could result in the women becoming more conscious of their

body image (*Grogan, 2008*). A meta-analysis revealed greater association for females and body dissatisfaction (*Weinberger et al., 2017*). Despite age or aging, this dissatisfaction was found to be very stable throughout adult female lives (*Tiggemann, 2004*). Thus, body dissatisfaction may play an important role in the impact of BMI on eating behavior in women. Compared to individuals with a normal weight, individuals with a high BMI showed greater body dissatisfaction among females (*Weinberger et al., 2017*). In African American girls (early adolescent), BMI was significantly associated with subsequent body dissatisfaction; BMI could also positively predict body dissatisfaction and body dissatisfaction as a mediate was supported on subsequent dieting behaviors (*Buckingham-Howes et al., 2018*). However, *Şanlier, Türközü & Toka (2016)* investigated the relationship between body image and food addiction in 793 students at a university in Ankara, Turkey, and the result showed a positive association between body weight dissatisfaction (BWD) and food addiction. According to the theory of *Heatherton & Baumeister (1991)*, people could show attention bias on immediate stimulus environments to escape from negative self-awareness, allowing them to avoid dealing with ego threatening information. This narrowing of attention results in disinhibition could affect the individual's attention bias (*Donofry et al., 2019*). Therefore, BWD may be a mediator of the relationship between BMI and food attention bias in women. BWD might explain why women with high BMIs pay more attention to food cues.

The aim of this study is to investigate the relationship between BMI and food attention bias, and the mediating role of BWD on this relationship in women. To test this, we adopt a dot-probe task with eye tracking to measure attention bias of food cues in women with different BMIs. According to the review of *Werthmann, Jansen & Roefs (2015)*, the attention by measuring response latencies is an indirect assessment, thus, visual attention can be examined directly through 'eye-tracking' by measuring eye-movements during the stimulus presentation. The reliability research for the food attention bias index also pointed out that eye tracking could produce more reliable food attention bias results (*Van Ens et al., 2019*).

In summary, we hypothesized that BMI would be associated with food attention bias of women, and there will be a mediating role for BWD on the association between BMI and food attention bias. We adopted the eye tracking study to examine this association. Response latency and eye tracking indicators (first fixation duration bias and direction bias) were the dependent variable indicators of attention bias. Response latency is the interval of time between presentation of a stimulus and detection of a response, first fixation duration bias is the initial maintenance of stimulus, direction bias is the initial attention orienting of stimulus, reflects attention vigilance (*Gao et al., 2012*).

## MATERIALS & METHODS

### Participants

We recruited 75 female college students aged 18 to 29 ($M_{age} = 21.99 \pm 2.53$) through advertisements placed in campus at Northwest Normal University. All participants were right-handed, with qualified vision. Exclusion criteria: (a) psychotic disorder, self-reported

history of psychological or psychiatric diagnose; (b) restraint eater, self-reported currently dieting or had experienced any problems with their eating; (c) substance abuse, smokers, alcohol users, self-reported current medication, smoke and alcohol uses. To control the influence of participants' expectations about the purpose of the study, we told them it is a survey about objects of daily life and cognition. All participants received a present (value: 25 RMB) after they finished the whole procedure. When participants finished the study, they were explained the purpose and asked them do not tell the purpose of the study to others, in case who interested in participating know the purpose. Table 1 shows the general characteristics of participants.

## Questionnaires

The *Negative Physical Self Scale-Fat (NPSS-F)* is the subscale of the Negative Physical Self Scale. The authors have permission to use this instrument from the copyright holders (*Chen, Jackson & Huang, 2006*). This subscale contains 11 items to assess the cognitive, affective, and behavioral facets of body weight dissatisfaction (BWD). The subscale is used for assessing body weight dissatisfaction among Chinese adolescents and adults. The items of the NPSS-F are presented as statements related to body weight. Example items include: "I am quite concerned about my weight" and "I am very distressed when I think about my weight". Participants rated their response on each item using a five-point Likert-type scale, ranging from 0 (never) to 4 (always). Total scores are calculated sum up the scores of all the items. Total scores ranged between 0 to 44. Higher scores mean more dissatisfaction with body weight, while lower scores mean less dissatisfaction with body weight (*Chen, Jackson & Huang, 2006*). The NPSS-F had internally consistent scores ($\alpha = .88$) as well as stable scores over 36 weeks ($r = .70$) among females and males from middle and high school students (*Chen & Jackson, 2007*). The reliability and validity of the NPSS-F also has been verified in North American females (*Ly, Misener & Libben, 2019*).

*Hunger*. Before the experiment, each participant was asked to report their current hunger (0 = extremely hungry, 100 = extremely not hungry).

*BMI*. Participant BMI was measured and recorded by asking about their height and weight which was then calculated to a BMI value.

## Materials

Food pictures in the experiment came from the experimental food picture library created by *Blechert et al. (2014)*. For the study, 10 high-calorie food pictures, 10 low-calorie food pictures and 40 neutral pictures (tools or instruments) were selected as experimental materials (20 as a filler materials). The high-calorie foods were different from the low-calorie foods on calorie ($p < .001$) and fat content ($p < .001$). In order to control the influence of the edible degree of food, all food pictures contained directly edible foods. All pictures were 400 × 400 pixels.

## Stimuli presentation

During the dot-probe task, all the pictures were divided into 30 pairs. In stimuli picture, each food picture (high-calorie or low-calorie) was paired with a non-food picture, for a

**Table 1  Basic characteristics of participants.**

|  | M | SD | Min | Max |
|---|---|---|---|---|
| Age | 22.23 | 2.87 | 18 | 29 |
| BMI (kg/m$^2$) | 21.39 | 3.12 | 15.94 | 33.38 |
| Hunger | 79.00 | 5.81 | 65 | 100 |
| BWD | 16.33 | 8.96 | 1 | 36 |

**Notes.**

M, Mean; SD, Standard deviation; BMI, Body Mass Index; BWD, Body Weight Dissatisfaction.

total of 20 pairs. Each filler picture consisted of two neutral pictures, for a total of 10 pairs. All picture pairs were matched as closely as possible on color and shape.

### Trial types

The dot-probe task consisted of 102 trials in total. In the task, each stimuli pair appeared four times, the filler pair appeared one time, a total 90 trials. A practice was conducted to familiarize participants with the task that included 12 trials. Each stimuli trial showed a food picture that was randomly present on the left or right side.

### Eye movement measurements

The Eyelink 1000 Plus (SR Research, Mississauga, Ontario, Canada) was utilized to record eye tracking indicators. The experiment only tracks the right eye, and the sampling rate is 1,000 Hz. The indexes for eye tracking data were direction bias and first fixation duration bias of food pictures (high-calorie and low-calorie) first fixation duration bias. Direction bias reflects an early attention distribution, and the first fixation duration bias reflects the continuity of the early attention distribution (*Field et al., 2004*; *Werthmann, Jansen & Roefs, 2015*). The effective fixation screening criteria are (*Applied Science Group, 2000*): (1) Participants' eyes are fixed on the "+" in the center of the screen before the stimulus appears; (2) the first saccade appears 100 ms after the stimulus appears; (3) when the stimulus appears, the first fixation occurred on a food or non-food stimulus.

### Attention bias scores

A response latency score was calculated by subtracting the mean response latency when the dot is consistent with the food picture from the mean response latency when the dot is inconsistent with the food picture. A positive score represents food attention bias, a negative score represents attention avoidance (*Field et al., 2004*).

A direction bias score was a percentage score indicating the proportion of trials in which the first fixation falls on the food picture to all trials in which first fixations fall within the interest area. Direction bias scores of more than 50% indicate early attention to food, and scores less than 50% represent early food attentional avoidance (*Castellanos et al., 2009*).

A first fixation duration bias score was calculated by subtracting the mean first fixation duration bias of non-food picture from the mean first fixation duration bias of the food picture. A positive score represents food attention bias, a negative score represents attentional avoidance (*Gao et al., 2011*).

## Designed and procedure

The current study was designed based on experimental research framework. Participants were asked not to eat any food 60 min prior to the experiment. Before participants were invited to enter the eye-tracking laboratory, they signed the informed consent and reported their current hunger, then they completed the food dot probe task. All tasks took about 20–30 min to complete. After the task, participants reported demographic information, and their height and weight measured by the research assistant.

## Dot probe task

Participants sat 60 cm from the monitor screen. Before the dot probe task, participants finished a standardized calibration procedure for eye tracking. Each trial began with the presentation of a black central fixation cross "+" for 500 ms, followed by a pair of pictures for 2,000 ms. The picture pair was presented with a picture on the left half of the screen and another on the right half. A black dot appeared randomly on the left or right side of the screen after picture. If the dot appeared on the left side of the screen, the participant pressed the "F" key on the keyboard, and if it appeared on the right side, they pressed the "J" key. When the participant pressed a key, the dot disappeared. Then, a blank screen was shown for 500 ms, followed by the next trial. When first block finished, participants could have a break with their head steady, then press any key to go into the second block. To evaluate the stability and the reliability of the dot-probe task, for each food picture (20 food pictures), by subtracting the two configurations in which food picture and dot position were consistent from those two configurations where they were inconsistent, an index indicating attention allocation toward food stimuli for each person, the bias index, was computed: [(food picture left and dot right + food picture right and dot left)−(food picture left and dot left + food picture right and dot right)]/2 (*Schmukle, 2005*). Cronbach's $\alpha$ for this task is 0.54.

## Statistical analysis

Data from the practice trials and filler trials used to balance the experiment were excluded. The response latency results were preprocessed, data with errors were removed (0.94% of data); we also removed trials with response latencies of more or less than 3SD (0.6% of data). Accuracy of all participants in the experiment was more than 90%. Eye tracking results were preprocessed, four participants whose fixation point was less than 80% in the area of interest were excluded. All data were analyzed utilizing the IBM SPSS v23.0. First, descriptive statistics to analyze the primary variables were run. In addition, Pearson correlations were examined for the primary variables (BMI, BWD, response latency, direction, duration). Then, a mediation model (model 4) of PROCESS Macro ver. 3.5 (*Hayes, 2018*) was used to examine the mediating effect of BWD on the relationship between BMI and food attention—the confidence intervals (95%) were generated by bootstrap with 5,000 re-samples.

## Ethics

Before the experiment, all participants gave informed written consent in accordance with the Code of Ethics of the World Medical Association (Declaration of Helsinki), and the

study protocol was approved by the Ethics Board of Northwest Normal University (ERB No. 2021045).

## RESULTS

### Preliminary analyses

Means, SDs, and bivariate correlations of primary variables (behavioral) are shown in Table 2. The results showed that, BMI was positively correlated with BWD ($r = .64$, $p < .001$) and the response latency of high-calorie foods was positively correlated with both BWD ($r = .35$, $p = .002$) and BMI ($r = .27$, $p = .02$). The response latency of low-calorie foods was no correlated with both BWD ($r = -.14$) and BMI ($r = .02$).

Means, SD, and bivariate correlations of the primary variables (eye movement) are shown in Table 3. The results showed that, BMI was positively associated with BWD ($r = .63$, $p < .001$) and first fixation duration bias of high-calorie foods was positively associated with both BWD ($r = .41$, $p < .001$) and BMI ($r = .27$, $p = .02$). The first fixation duration bias of low-calorie foods was no correlated with both BWD ($r = .18$) and BMI ($r = .19$). The direction bias of high-calorie foods was no correlated with both BWD ($r = -.08$) and BMI ($r = -.07$), and the direction bias of low-calorie foods was no correlated with both BWD ($r = .2$) and BMI ($r = .05$).

### Testing for mediation

Based on the correlation results, only response latency of high-calorie food and first fixation duration bias of high-calorie food could into testing for mediation, there were two hypotheses: (1) BWD could mediate the association between BMI and response latency of high-calorie food, and (2) BWD could mediate the association between BMI and first fixation duration bias of high-calorie food. Table 4 showed the effect of BMI on response latency of high-calorie food, the effect of BMI on BWD as well as the mediation effect of BWD between BMI and response latency of high-calorie food.

Table 4 shows the findings regarding the direct association between BMI and response latency of high-calorie food and the mediating role of BWD them. Bootstrap analysis showed the indirect effect of BMI on response latency of high-calorie food through BWD ($B = 3.15$, 95% CI $= [.89, 5.69]$). The total effect of BMI on response latency of high-calorie food was $B = 4.23$, 95% CI $= [.63, 7.82]$. In model 2 (the direct association between BMI and response latency of high-calorie food and the mediating role of BWD them) of Table 4, the effect of BMI on response latency of high-calorie food was $B = 1.08$, 95% CI $= [-3.52, 5.67]$, but the effect of BMI on BWD was $B = 1.85$, 95% CI $= [1.33, 2.36]$ (model 1, the direct association between BMI and BWD), the effect of BWD on response latency of high-calorie food was $B = 1.70$, 95% CI $= [.11, 3.30]$. Therefore, the relationship between BMI and response latency of high-calorie food was a complete mediation of BWD (see Fig. 1).

Table 5 showed the effect of BMI on first fixation duration bias of high-calorie food, the effect of BMI on BWD, and the mediation effect of BWD between BMI and first fixation duration bias of high-calorie food.

**Table 2  Descriptive statistics and correlations among variables (behavioral) ($N = 75$).**

| Variables | M | SD | 1 | 2 | 3 | 4 |
|---|---|---|---|---|---|---|
| 1. BWD | 16.33 | 8.87 | 1.00 | | | |
| 2. BMI | 21.39 | 3.12 | .64*** | 1.00 | | |
| 3. Response (H) | 14.98 | 49.84 | .35** | .27** | 1.00 | |
| 4. Response (L) | −10.26 | 46.86 | −.14 | .02 | .46*** | 1.00 |

Notes.

M, mean; SD, standard deviation; BWD, body weight dissatisfaction; BMI, body mass index; Response (H), Response latency of high-calorie food; Response (L), Response latency of low-calorie food.

*$p < .05$
**$p < .01$
***$p < .001$

**Table 3  Descriptive statistics and correlations among variables (eye movement) ($N = 71$).**

| Variables | M | SD | 1 | 2 | 3 | 4 | 5 | 6 |
|---|---|---|---|---|---|---|---|---|
| 1. BWD | 16.35 | 8.92 | 1.00 | | | | | |
| 2. BMI | 21.39 | 2.99 | .63*** | 1.00 | | | | |
| 3. Duration (H) | 85.31 | 191.08 | .41*** | .27* | 1.00 | | | |
| 4. Duration (L) | 24.19 | 182.41 | .18 | .19 | .73*** | 1.00 | | |
| 5. Direction (H) | .52 | .08 | −.08 | −.07 | .001 | .14 | 1.00 | |
| 6. Direction (L) | .53 | .07 | .20 | .05 | .15 | .13 | .20 | 1.00 |

Notes.

M, mean; SD, standard deviation; BWD, body weight dissatisfaction; BMI, body mass index; Duration (H), First fixation duration bias of high-calorie food; Duration (L), First fixation duration bias of low-calorie food; Direction (H), Direction bias of high-calorie food; Direction (L), Direction bias of low-calorie food.

*$p < 0.05$
**$p < 0.01$
***$p < .001$

**Table 4  The mediation effect of BWD in the relationship between BMI and response latency of high-calorie food ($N = 75$).**

| Antecedent | Model 1 BWD | | | | | Model 2 Response (H) | | | | |
|---|---|---|---|---|---|---|---|---|---|---|
| | B | 95% CI | | SE | t | B | 95% CI | | SE | t |
| | | Lower bound | Upper Bound | | | | Lower bound | Upper Bound | | |
| BMI | 1.85 | 1.34 | 2.37 | .25 | 7.19*** | 1.08 | −3.52 | 5.67 | 2.30 | .47 |
| BWD | | | | | | 1.70 | .11 | 3.30 | .80 | 2.12* |
| $R^2$ | | .41 | | | | | .13 | | | |
| F | | 51.76*** | | | | | 5.14** | | | |

Notes.

SE, standard error; BMI, body mass index; BWD, body weight dissatisfaction.

*$p < .05$
**$p < .01$
***$p < .001$

Table 4 shows the findings regarding the direct association between BMI and first fixation duration bias of high-calorie food and the mediating role of BWD them. Bootstrap analysis showed an indirect effect of BMI on first fixation duration bias of high-calorie food through BWD ($B = 15.83$, 95% CI $= [6.33, 32.41]$). The total effect of BMI on first fixation duration bias of high-calorie food was $B = 17.37$, 95% CI $= [2.58, 32.15]$. In model

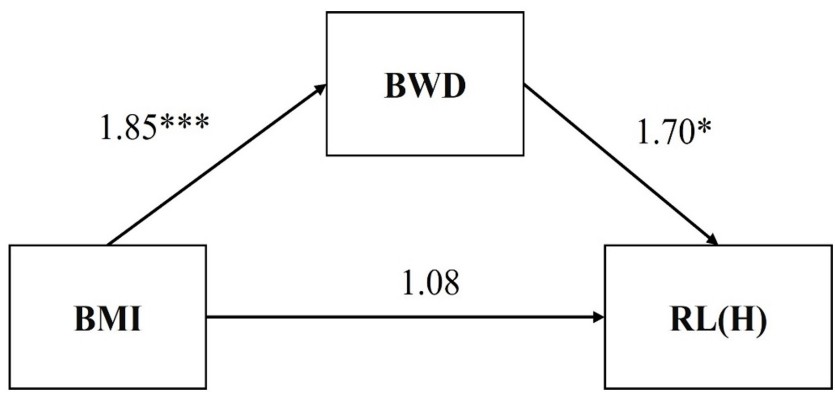

**Figure 1** **Mediation model of hypotheses 1.** An asterisk (*) indicates $p < .05$, three asterisks (***) indicate $p < .001$.

**Table 5** **The mediation effect of BWD in the relationship between BMI and first fixation duration bias of high-calorie food ($N = 71$).**

| Antecedent | Model 3 BWD | | | | | Model 4 Duration (H) | | | | |
|---|---|---|---|---|---|---|---|---|---|---|
| | B | 95% CI | | SE | t | B | 95% CI | | SE | t |
| | | Lower bound | Upper Bound | | | | Lower bound | Upper Bound | | |
| BMI | 1.89 | 1.34 | 2.345 | .28 | 6.69*** | 1.54 | −16.72 | 19.80 | 9.15 | .17 |
| BWD | | | | | | 8.37 | 2.26 | 14.49 | 3.06 | 2.73** |
| $R^2$ | | | .40 | | | | | .17 | | |
| F | | | 46.10*** | | | | | 6.74** | | |

Notes.

SE, standard error; BMI, Body Mass Index; BWD, Body Weight Dissatisfaction; Duration (H), First fixation duration bias of high-calorie food.

*$p < .05$
**$p < .01$
***$p < .001$

4 (the findings regarding the direct association between BMI and first fixation duration bias of high-calorie food and the mediating role of BWD them) of Table 5, the effect of BMI on first fixation duration bias of high-calorie food was $B = 1.54$, 95% CI = $[−16.72, 19.8]$, but the effect of BMI on BWD was $B = 1.89$, 95% CI = $[1.33, 2.45]$ (model 3, the direct association between BMI and BWD), the effect of BWD on first fixation duration bias of high-calorie food was $B = 8.37$, 95% CI = $[2.26, 12.49]$. Therefore, the relationship between BMI and first fixation duration bias of high-calorie food was a complete mediation of BWD (see Fig. 2).

## DISCUSSION

In the present study, we examined the relationship between BMI and food attention bias of women and the mediation effect of BWD. The results showed that BMI could positively predict the response latency and first fixation duration bias of food for women and that there was a complete mediating role for BWD on the association between BMI and food attention bias.

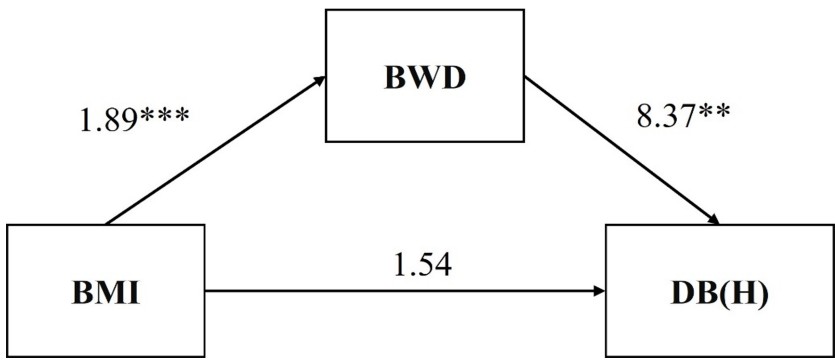

**Figure 2** **Mediation model of hypotheses 2.** Two asterisks (**) indicate $p < .01$, three asterisks (***) indicate $p < .001$.

The result of the relationship between BMI and food attention supports the study of *Hendrikse et al. (2015)*; they analyzed the results of 19 studies on attentional bias in individuals with obesity, including 3 studies containing an adolescent sample and 16 studies containing an adult sample. Their findings showed that aside from 4, 15 studies supported the notion of enhanced reactivity to food cues in overweight and obese individuals. According to the incentive sensitization model (*Robinson & Berridge, 1993*), the reward of food would increase attention bias toward salient cues. The greater connectivity between brainstem regions and reward regions is consistent in individuals with a high BMI who had a food addiction (*Ravichandran et al., 2021*). For people with obesity, the pairing of food cues and reward of food often appear to result in hypersensitization of the reward-processing system, further leading to preferential attention to highly palatable food (*Hagan et al., 2020*). The present study showed, for women, the higher the BMI, the more likely they are to pay attention to high-calorie foods. A previous study that also adopted a visual probe task to evaluate attention bias towards unhealthy foods at different stages of attention processing in a severe obesity and binge eating sample, found that all participants (high BMI with binge eating and high BMI with no binge eating) had a positive attention bias to food. However, this positive attention bias to food was more pronounced in the binge eating group (severe obesity). This result helps to further confirm that individuals with high BMI could pay more attention to unhealthy food (*Deluchi et al., 2017*). With this, continued selective attention to food would facilitate food intake (*Field & Cox, 2008*). One theory for this selective attention is that for individuals with obesity it feels more rewarding to eat high-calorie foods than low-calorie foods (*Stice et al., 2009*), and such food-related cues would then activate the dopaminergic reward pathways (*Twieg, Knowlton & Cox, 2008*). This activation of the dopamine pathway increases the attention of food-related cues in the environment (*Smith et al., 2020*; *Werthmann et al., 2011*).

Moreover, the present study found the complete mediation effect between BMI and food attention bias in women. It manifests that one possible effect of BMI to attention bias of high-calorie foods is through BWD in women. Previous studies found a strong association between BMI and disordered eating symptoms (*Buckingham-Howes et al.,*

*2018*; *Xu et al., 2018*). BWD as a negative self-awareness would change attention bias of people to an immediate stimulus environment (food cues) because overeating is a kind of strategy to escape this negative status, allowing them to avoid dealing with ego threatening information (*Evers, Stok & De Ridder, 2010*; *Heatherton & Baumeister, 1991*). The narrowing of attention results in disinhibition which could affect the individual's attention bias and eating behavior for food (*Czepczor-Bernat & Brytek-Matera, 2020*; *Donofry et al., 2019*). BWD occurs when weight standards whether they are self-selected or imposed by society are inconsistent with one's current weight. For women, it may that body-related information from mass media influences such standards or BWD. The present result and this statement are inconsistent with the study by *Gao et al. (2014)*, which suggested that body dissatisfaction did not increase after exposure to thin or fat body images. However, in another study on body dissatisfaction, researchers adopted a baseline survey and modified dot-probe task to explore the effect of experiences of body dissatisfaction in daily life using attention bias for body shape and weight information. The results did suggest that short-term attention biases of appearance-related information could be a vulnerability factor for the prolonged persistence of body dissatisfaction experiences in daily life (*Fuller-Tyszkiewicz et al., 2020*). The present study is also consistent with the emotional eating theory. *Bruch (1973)* suggested that negative emotions lead to increased eating (*i.e.*, emotional eating). In emotional eating, increased food intake is a kind of compensatory behavior to decreases negative emotion (*Young & Limbers, 2017*). Based on these studies and the present study's results, women with high BMIs may experience more dissatisfaction with their body weight; to decrease such feelings they turn to eating more satisfying or high-caloric foods.

In the present study, we found the food attention bias on response latency and eye tracking, but for direction bias compared to duration, there was a weaker association between it and BMI. This is consistent with the reliability of attention bias for food for the study. The results showed that direction bias had poor internal and test-retest reliability, and first fixation duration bias had excellent internal and acceptable test-retest reliability; response latency also had acceptable internal and good test-retest reliability (*Van Ens et al., 2019*).

Despite that attention bias for food can predict food intake, it could still be changed by other factors. A previous study suggested that feminine norms effect food intake for women. The study reported that women are encouraged to regulate their behaviors, including their health-related behaviors and found that enhanced conformity to the feminine norm of modesty is associated with decreased food consumption (*Le, 2019*). that the present result worked only in women, as women were more likely to feel body weight dissatisfaction than men. Another survey found that among women and girls, dissatisfaction with the body is usually related to weight, especially the desire to lose weight, while among men and boys, dissatisfaction is usually related to concerns about not being thin enough or having underdeveloped muscles (*Grogan, 2008*). The relationship between BMI and food attention in men needs further exploration. According to the present study, further work can focus on interventions of body dissatisfaction, for example, how attention bias modification training could change food intake in overweight and obese women (*Smith et al., 2020*).

Besides, considering the interaction between BMI and attention bias of food (*Hagan et al., 2020*), the present study merely provides a possibility that women with high BMI showed more attention bias to food is attributable to high BMI inducing more BWD. Further study can explore the relationship among these factors deeply.

### Limitation

Although the investigation of the mediating role of BWD on the association between BMI and food attention bias in women through eye tracking were the key strengths of this research, a limitation still should be noted. Attention bias not only could predict subsequent food intake, type of intake, etc., but it could also indicate an emotional ambivalence or approach-avoidance pattern towards food cues (*Deluchi et al., 2017*). The various phases of attention have different attention patterns which represent specific mental activity (*Koster et al., 2005*), which our result may not show. Additionally, *Meneguzzo et al. (2021)* found that there is significant relationship between sexual orientation add BWD. This may limit our conclusions' generalizability. Future studies could systematically examine the effect of socio-cultural and psychological factors on BWD.

## CONCLUSIONS

In general, our study examined the relationship between BMI and food attention bias, where women with a higher BMI paid more attention to food cues. Furthermore, we found the mediating functions between BMI and food attention bias, which provides a possible explanation for the effect of BMI. BMI could positively predict the BWD of women and BWD could positively predict food attention bias. In women, gaining weight can lead to a more negative body related self-evaluation, which as a result could make them pay more attention to food to escape this status. Overall, the present investigation highlights the role of BWD on the association between BMI and food attention bias in women.

### Funding

This research was supported by the research fund of the National Natural Science Foundation of China No. 32160202. The funders had no role in study design, data collection and analysis, decision to publish, or preparation of the manuscript.

### Grant Disclosures

The following grant information was disclosed by the authors:
National Natural Science Foundation of China: 32160202.

### Competing Interests

The authors declare there are no competing interests.

### Author Contributions

- Aibao Zhou conceived and designed the experiments, authored or reviewed drafts of the article, and approved the final draft.

- Pei Xie conceived and designed the experiments, performed the experiments, analyzed the data, prepared figures and/or tables, and approved the final draft.
- Md Zahir Ahmed conceived and designed the experiments, performed the experiments, analyzed the data, prepared figures and/or tables, authored or reviewed drafts of the article, and approved the final draft.
- Mary C. Jobe conceived and designed the experiments, authored or reviewed drafts of the article, and approved the final draft.
- Oli Ahmed analyzed the data, authored or reviewed drafts of the article, and approved the final draft.

## Human Ethics

The following information was supplied relating to ethical approvals (i.e., approving body and any reference numbers):

The Ethics Board of Northwest Normal University approved the study (ERB No. 2021045).

## Data Availability

Raw data is available in the Supplemental Files and Figshare:

Xie, Pei (2021): DATA.sav. figshare. Dataset. https://doi.org/10.6084/m9.figshare.17469461.v3

## Supplemental Information

Supplemental information for this article can be found online at http://dx.doi.org/10.7717/peerj.13863#supplemental-information.

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
