# Peer review of "Body mass index and attention bias of food cues in women: a mediation model of body weight dissatisfaction"

_PeerJ, doi:10.7717/peerj.13863_

## Round 0.1 · original submission · Major Revisions

· Academic Editor

Major Revisions

Dear authors,

Please reply point by point to reviewers' comments and follow their suggestions carefully.

·

Basic reporting

This short article aims to expose the mediating effect of body weight dissatisfaction on the link between the body mass index and food attention bias (both response latency and duration bias). The idea of the article is easy to grasp and the synthetic writing makes the article pleasant to read. Yet, there are a few major things that needs to be addressed before being eligible for a potential publication.

1) The use of a cross-sectional design to address a causal question means that the conclusions (BMI causes food attention bias) cannot be backed up. Although I can conceive that it is impossible to modulate BMI within participants, authors should be a few times more parsimonious in writing their results and the scope of the model. Thus, on all the presented links, the independent variable may be the dependent variable and vice versa. For example, it is likely that obesity modulates the attentional effect of food, but that it is the individual attentional differences on food that leads some people to be obese.

2) It was unclear to me what happened to the attention bias related to low-calorie food cues. In the “eye measurement” section, it is written that “indexes for eye tracking data were direction bias and duration bias of high-calorie foods and low-calorie foods”. This sentence is ambiguous. I believe what the authors mean is that these 2 biases were measured for both high- and low-calorie foods. However, the authors do not refer at all to the results obtained on the low-calorie condition and do not describe the bias in opposition to that obtained in the high-calorie condition.

3) Similarly, the authors speak of two different models on latency bias as well as two different models on durations bias. After rereading several times, I do not understand what was inserted in the two different models.

4) Exclusion criteria: (a) psychotic disorder; (b) restraint eater; (c) substance abuse. How did the authors measured that ?

5) Were participants rewarded ?

6) It would be appropriate to also insert the direct effects between X and Y on the mediation graphs

7) English is not my mother tongue, nonetheless I found a lot of wording which seems strange to me. I believe that a proofreading of the article would be appropriate. Here is a non-exhaustive list :
line 43 : “A recent meta-analysis by Hagan et al. (2020) was conducted a dot-probe”
Line 219 : “BMI was positively associated BWD (r = .64, p < .001) and the response latency of high-calorie foods was positively associated both BWD (r = .35, p = .002) 221 and BMI (r = .27, p = .02). –> “With” was forgotten twice.
Line 241 : “Bootstrapping analysis showed that the indirect effect of BMI on response latency of high calorie food through BWD”
Line 311 ; However (or Nonetheless) instead of But
Line 525 ; Skip a line

Experimental design

No comment

Validity of the findings

No comment

Additional comments

No comment

Reviewer 2 ·

Basic reporting

I offer my compliments to the authors for this interesting and important manuscript. I have a few comments in an attempt to clarify or add some missing information.

Title and abstract

1. I suggest describing the study design in the title or in the abstract once although it is implicit, this information is not described anywhere in the manuscript.

2. Please, provide the age range or the mean age (or both) of the study’s participants.

Introduction

3. Line 38: when mentioning attention bias to food cues, I suggest adding a brief description of what it is (or, at least, what attention bias is). It would be helpful for those readers who are not familiar with this expression.

4. Lines 53-54: again, when mentioning top-down attention bias, I suggest adding a brief description of what it is.

5. Lines 68-69: although the mass media is the main sociocultural agent influencing women’s internalization of appearance ideals, current sociocultural theories of body image suggest that both media, family members, peers, and significant others have a strong influence on thin-ideal internalization. I suggest rewriting this part of the text to consider a more broad scenario.

6. Lines 90-31: before the sentence “To test this (...), I suggest describing the aim of the study.

Experimental design

Material and methods
7. Please, provide the participants' age range.

8. Please, provide information on how was the recruitment process. Also, what was told about the research to the participants? Which information did they receive to participate?

9. Please, provide information on the study design.

10. When describing the Negative Physical Self Scale-Fat (NPSS-F), please, provide information on how its score is calculated.

11. The tables sent in separate files are without titles and subtitles. Also, tables and figures must be mentioned in the text.

12. Regarding the calculation of the duration bias score, should not it be the opposite?

13. Regarding participants' height and weight measures, it is not clear if these measures were self-reported or not: “After the task, participants reported demographic information and their measured height and weight”. They reported their “measured height and weight”? It is not clear. Anyway, if these measures were self-reported, although they are considered valid for epidemiological purposes, I was wondering if in the particular case of studies like this it would not be biased, even if reported before the test (depending on the information about the test the participants received), or (maybe worse) after the test (after seeing pictures of food).

14. The values of BWD should not be the same in Table 2 and Table 3? And the same for the correlation between BMI and BWD? Also, could not Table 2 and Table 3 be grouped?

Validity of the findings

No comment.

Additional comments

No comment.

Reviewer 3 ·

Basic reporting

The paper covers an interesting and actual topic. The manuscript is well written and it is clear. The structure of the paper is well organized.

Experimental design

I have some concerns about the statistical analysis.
-The recent literature has shown the need to evaluate the stability and the reliability of the dot-probe task, so please include these analyses in your paper.
- The recent literature has shown that BWD is linked to different features than sexes, like sexual orientations (see https://doi.org/10.1007/s40519-020-01047-7). Have you considered this element?
- In your sample, there are participants with BMIs very low (15 kg/m2). This BMI is very suggestive of anorexia nervosa. Have you evaluated this aspect? How have you evaluated psychiatric conditions?
- Previous dot-probe papers with food images have pointed out the effects of colors and shape differences between foods. Have you considered these aspects when you have structured your task?

Validity of the findings

The conclusions are supported by the data reported and the results could have relevance in the literature. However, I think the authors should address my comments regards methodology.

Additional comments

- Please include captions and titles to tables and figures

---

## Round 0.2 · accepted · Accept

· Academic Editor

Accept

Dear authors,

Congratulations on your manuscript!

·

Basic reporting

No issues

Experimental design

No issues

Validity of the findings

issues clearly answered by authors

Additional comments

The authors responded to the issues highlighted with relevance.

I have no more changes nor any concern to expose.

The article is now of sufficient quality for submission to the journal.

Thank you for allowing me to review this article.

Reviewer 3 ·

Basic reporting

I think the authors have addressed all my concerns. The paper is now more clear as regards results and discussion with the previous literature data

Experimental design

The authors have addressed my concerns about the validity of the task.

Validity of the findings

The authors have increased the description of the methods for future replications, as well as the limits.